# Palladium–Copper Bimetallic Aerogel as New Modifier for Highly Sensitive Determination of Bisphenol A in Real Samples

**DOI:** 10.3390/ma16186081

**Published:** 2023-09-05

**Authors:** Zehao Fang, Junyan Wang, Yilei Xue, Mozhgan Khorasani Motlagh, Meissam Noroozifar, Heinz-Bernhard Kraatz

**Affiliations:** 1Department Physical and Environmental Sciences, University of Toronto Scarborough, 1265 Military Trail, Toronto, ON M1C 1A4, Canada; zehao.fang@mail.utoronto.ca (Z.F.); junyansteven.wang@mail.utoronto.ca (J.W.); yilei.xue@mail.utoronto.ca (Y.X.);; 2Department of Chemistry, University of Toronto, 280 St. George Street, Toronto, ON M5S 3H6, Canada

**Keywords:** bimetallic palladium–copper aerogel, bisphenol A, electrocatalytic, modified graphite paste electrode, real samples

## Abstract

In this study, a bimetallic palladium–copper aerogel was synthesized and used for modification of a graphite paste electrode (Pd-Cu/GPE), allowing the sensitive determination of bisphenol A (BPA). Different techniques, such as SEM, TEM, XPS, and AFM, were used for characterization of the Pd-Cu aerogel. To elucidate the properties of the Pd-Cu/GPE, the electrochemistry methods such as differential pulse voltammetry (DPV) and electrochemical impedance spectroscopy were used. DPV measurements were conducted in phosphate electrolyte and buffer solution (0.2 M PBS, pH 5) at a potential range from 0.4 to 0.9 V vs. Ag/AgCl. The DPVs peaks currents increased linearly with BPA concentrations in the 0.04–85 and 85–305 µM ranges, with a limit of detection of 20 nM. The modified electrode was successfully used in real samples to determine BPA, and the results were compared to the standard HPLC method. The results showed that the Pd-Cu/GPE had good selectivity, stability, and sensitivity for BPA determination.

## 1. Introduction

Bisphenol A (BPA) is a phenolic monomer compound with a wide application as raw material in plastic and epoxy resin industrial manufacturing [1,2,3,4]. Its extensive usage directly led to an annual release of roughly 2000 tons of BPA into the environment on a global scale. BPA is reported to be a hazardous pollutant due to its negative effects on humans, aquatic life, and wildlife [4,5]. As a xenoestrogen and an endocrine-disrupting chemical, BPA can interact with various biological receptors, causing abnormal hormone responses and endocrine disorders in the hypothalamus–pituitary–gonadal glands and the pituitary–adrenal function [6,7,8] Prostate and breast cancer, infertility, obesity, and diabetes are feasible under intensive exposure to BPA [9,10]. The most important exposure to BPA is through dietary routes, occupying more than 90% of the total exposure to BPA [10,11,12]. Even at low concentrations, exposure to BPA through food and drinking water is carcinogenic and can cause irreversible damage to the organs or tissues of animals and human bodies [13]. The maximum concentration levels of BPA in drinking water established by the European Union, China, and Japan are 2.5 µg/L (11 nM), 10.0 µg/L (44 nM), and 100 µg/L (440 nM), respectively [14]. The safe and tolerable dose reported by the United States Food and Drug Administration is 50 μg/kg/day, and any intake above this level is risky [12]. As a result, the use of BPA is prohibited in a number of countries. Canada, as the first country to list BPA as toxic, concluded that BPA is risky to the environment and human health and committed to reducing its concentrations in the environment from 2010 onwards [2]. However, BPA cannot be treated or degraded by the existing wastewater treatment plants; therefore, timely detecting of BPA and environmental monitoring of BPA are highly valued as limiting the release of BPA becomes a global concern [13].

A variety of detection methods have been developed to effectively detect BPA in water and food [9]. Due to the tremendously low concentration of BPA in the environment and environmental matrix complexities, the detection methods of BPA need to be extremely sensitive and selective. The most popular methods are represented by combinations of chromatography and mass spectrometry [15,16]. Among those methods, LC-MS is most outstanding due to its sensitivity and accuracy in the quantitative determination of BPA [15,17]. A low limit of quantitation (LOQ) for BPA is reported to be 0.03 ng/mL, implying the great performance of LC in detecting BPA [18]. Despite high sensitivity and selectivity, these chromatography- and mass spectrometry-based methods have their common shortage in the cost, analytical time, and complicated pretreatment of samples [16]. Electrochemical detection has attracted significant attention due to its low cost, effectiveness, simple operation, and speed, showing great potential in detecting BPA more conveniently than current methodologies [16]. Electrochemical detection monitors the redox status of the analytes under a wide range of potentials and, hence, quantifies the analytes by the current changes. BPA has two electroactive phenolic hydroxyls and, as a result, has an edge to be detected by electrochemical methods [19]. As nanotechnology has been developing rapidly in recent years, electrochemical detection is amplified by nanocomposite modifiers and shows greater sensitivity and selectivity [13]. Different strategies were reported to be utilized in working electrode preparation, including Au, Pt, and glassy carbon. However, most of the working electrodes face shortcomings in that they are expensive, and it is difficult to renew their surfaces. Herein, we make use of graphite paste electrodes as working electrodes due to their low cost, ease of preparation, and wide potential range. In addition, graphite paste electrodes are reported to be non-poisonous and have a porous surface and low residual current, and their surface is readily renewed [20,21].

Pd has been employed in a variety of catalytic applications for fuel electrooxidation [22,23]. Copper nanoparticles have been considered outstanding electrochemical catalysts [24], and the copper content of the alloy has been examined for electrooxidation reaction [25]. Also, addition of the Pd is expected to improve the reducibility of copper and enrich the catalyst surface with electrons, hence increasing the electron transferring rate [26,27]. The existing research on Pd-Cu bimetallic nanoparticles is mainly focused on its catalytical capability for the electroreduction of CO_2_ and NO [26,27]. According to this research, bimetallic Pd-Cu nanoparticles catalyst has the potential advantage of providing binding site variety, which boosts catalytic activity and has remarkable potential in the electrocatalysts research [28]. Mo et al. reported the application of the first bimetallic Au-Pd incorporated in carboxylic multi-walled carbon nanotubes as the supporter to improve electron transport of the poly (diallyldimethylammonium chloride), and used for electrochemical determination of BPA [29]. In this study, different types of metallic aerogels—Pd, Cu, and bimetallic aerogel Pd-Cu with three different ratios, 5:1, 3:3, and 1:5—were prepared. The mentioned aerogels were used for modification of the graphite paste electrode (GPE), and their electrocatalytic capabilities for BPA detection were investigated. In this contribution, we describe the results of our studies on the use of Cu-Pd aerogel for sensitive detection of BPA in the real samples. And, in a side-by-side comparison to the standard technique, we hope to demonstrate the utility of our approach for BPA detection electrochemically.

## 2. Materials and Methods

### 2.1. Materials and Solutions

Bisphenol (BPA, 97%), graphite powder (G, <20 micron), Paraffin oil, copper chloride (99.0%), palladium (II) chloride (99%), potassium hydroxide, phosphoric acid (85%), sodium carbonate, glyoxylic acid, and sodium borohydride were all obtained from Sigma-Aldrich Company (Oakville, ON, Canada). Water purification systems, such as the Pascada LS water purification system, manufactured by Pall Co., Mississauga, ON, Canada, were employed in the preparation of all solutions. Phosphoric acid electrolyte and buffer solution (PBS) with pH in the range of 2.0 to 8.0 was prepared using 0.2 M phosphoric acid and adding 5.0 M NaOH. The 0.005 M BPA was prepared by dissolving it in 100 µL of 1 M KOH and diluting it with Milli-Q water. The PdCl_2_ was dissolved in hydrochloric acid to make H_2_PdCl_4_ precursor solution.

### 2.2. Synthesis of Pd-Cu Aerogel

The Cu-Pd aerogels were prepared using a procedure as demonstrated in previous works [30,31]. In a glass vial, a fresh aqueous solution of glyoxylic acid (100 mg) and Na_2_CO_3_ (500 mg) was mixed into a 10 mL solution of H_2_PdCl_4_ (5 mM, by dissolving 8.9 mg PdCl_2_ in 10 mL HCl 100 mM) and CuCl_2_ (1.0 mM, 1.34 mg CuCl_2_) and then sonicated for 10 min. Next, the glass vial containing the suspension was transferred into an oven at 70 °C for 1 h to obtain a cloudy dark-grey-color solution. To complete the reduction, NaBH_4_ (35 mg) was added after the solution had been cooled to 40 °C, and the reaction mixture was left standing for another 3 h to cool and precipitate all black solid products. The resulting solid was washed sequentially with distilled water, ethanol, and acetone (3 times, with 200 mL each solvents), followed by overnight freeze-drying to obtain porous Pd-Cu aerogel with the mole ratio 5:1. By contrast, different mole ratios of the precursor salts of Pd-Cu (5:0, 2.5:2.5, 1:5, 0:5) were prepared via the same process.

### 2.3. Preparation of Modified Electrode

A mixture of 1.0 mg Pd-Cu aerogel and 199 mg graphite was ground for 10 min with a mortar and pestle, resulting in a homogeneous mixture. Three drops of paraffin oil were added to this mixture, following by further mixing for 15 min. This paste was placed into a 2.0 mm diameter glass tube, and copper wire was used as an electrical contact. This electrode was denotated as Pd-Cu/GPE. A similar method was used for preparations of modified GPE that only contained Cu and Pd aerogels and were denoted as Cu/GPE and Pd/GPE, respectively. A bare GPE was prepared using 200 mg graphite powder.

### 2.4. Instrumentation

All electrochemical measurements were carried out at room temperature using an Autolab Potentiostat/Galvanostat (PGSTAT 302 N Metrohm AG, Herisau, Switzerland) controlled by NOVATM 2.1.2 software (Metrohm AG, Herisau, Switzerland). The three-electrode system used for CV measurements included a glassy carbon electrode (GCE) as the working electrode (diameter 2 mm), a reference electrode (saturated Ag/AgCl, 3 M KCl), and a platinum wire as the counter electrode. EIS data were collected with an oscillation amplitude of 0.01 V and a frequency range of 100 kHz to 0.1 Hz. The morphology of the nanocomposite was examined using a Hitachi S-530 scanning electron microscope (SEM) (Hitachi, Chiyoda, Tokyo, Japan). The transmission electron microscopy (TEM) images were collected by a Hitachi H-7500 transmission electron microscope (Hitachi, Chiyoda, Tokyo, Japan). X-ray photoelectron spectroscopy (XPS) measurements were obtained using the Thermo Fisher Scientific K-Alpha XPS spectrometer (Thermo Fisher Scientific-E, Grinstead, UK). Survey spectra (nominal 400 μm spot, 200 eV pass energy (PE), 1 eV step size) were obtained for all samples, followed by low-resolution spectra (150 eV pass energy, 0.2 eV step size) on the spectral regions of interest (Pd 3d, Cu 2p), from which the composition was obtained. All data processing was performed using the software supplied with the system (Avantage 5.957). A JPK NanoWizard^®^ 4 (JPK Instruments, Berlin, Germany) coupled with an inverted optical fluorescent microscope (Zeiss Axio Observer 7, Jena, Germany) was utilized to image Pd-Cu aerogel via the Quantitative Imaging (QI) modality. An SNL-10A probe with the spring constant equaling to 0.35 N·m^−1^ and a 2 nm tip radius was selected to conduct imaging. A Labconco freeze dryer was used for aerogels drying. A VWR SB70P pH meter was used to conduct pH measurements when creating buffer solutions. An 18.2 MΩ cm Millipore Milli-Q water purification system was applied to provide purified deionized water. The analytical performance of the modified graphite paste electrode was also compared with a standard method, UNE-EN: 13130-13:2005, using high-performance liquid chromatography (HPLC) [32]. For this, a Varian high-performance liquid chromatography (HPLC) system equipped with a variable UV-vis detector, Varian Model 320, and a six-port injector valve Rheodyne, including a 20 μL stainless steel sample loop, was used. Chromatographic separations were developed on a Phenomenex C_18_ column (5 μm, 4.6 × 250 mm) under an isocratic program, using 55% of acetonitrile (solvent A) and 45% of water (solvent B) with the flow rate 0.7 mL·min^−1^ as mobile-phase components. Data analysis was carried out using ProStar software. Disposable membrane filters of 0.22 μm pore size (Millipore Corp., Milford, MA, USA) were used for filtration of the real samples. Analytical balance Metter Toledo model AB54-S/fact and VWR sonicator, Model 50D, Switzerland, were employed for weighing and sonication, respectively.

## 3. Results and Discussion

### 3.1. Characterization

SEM was used to characterize the morphology of the Pd-Cu aerogel, as illustrated in Figure 1a–c. The Pd−Cu aerogel displays a superstructure with a high porosity, as well as extended nanochains. Moreover, these images also show 3D network-like structures with numerous open, interconnected pores and tunnels.

The XPS survey of Cu-Pd (see Figure 1d) confirmed the presence of Pd and Cu. The Pd ^3^d_5/2_, Pd ^3^d_3/2_, Cu ^2^P_3/2_, and Cu ^2^P_1/2_ binding energies were observed at 338.1, 532.1, 932.1, and 951.8 eV, respectively. Due to the association of Pd and Cu in the formation of the bimetallic Pd-Cu, binding energies were shifted slightly higher than the pure forms of both Pd ^3^d_5/2_ (335.5 eV) [33] and Cu ^2^P_3/2_ (932.8 eV) [34]. The TEM micrographs for Pd-Cu aerogel are shown in Figure 1e,f. The TEM micrographs display the connection of primary NPs to construct the 3D nanoarchitectures with plentiful open, interconnected pores and tunnels. Furthermore, the surface topography of Pd-Cu aerogel was imaged using AFM and is shown in Figure 1g,h. The AFM topographic images confirmed a branching morphology and nanopores.

### 3.2. Analytical Performance of the Modified Electrodes

The electrochemical oxidation of BPA was tested in 0.2 M PBS at pH 5.0 using various types of electrodes with the differential pulse voltammetry technique (DPV) to compare their performance in detecting the BPA, and the results are shown in Figure 2. The final modified electrode containing Pd-Cu/GPE was compared to three other electrodes, namely, bare GPE, Pd/GPE, and Cu/GPE. As demonstrated in the DPV overlay in Figure 2, Pd-Cu/GPE displayed a sharp current signal for BPA at 0.612 V, with a higher current intensity for the BPA. The bare GPE displayed low and broad peaks at 0.658 V, which suggested that the electron transfer kinetics was significantly slow. Both Pd/GPE and Cu/GPE displayed weak current peaks at 0.624 and 0.621 V, respectively, that remained relatively low and very broad. Additionally, Pd-Cu/GPE and others electrodes had extra background peaks that appeared due to the electrode itself, which was present in the blank buffer signal.

The Nyquist plots of the bare GPE and of the Pd-Cu/GPE are shown in Figure 2b, in which the electrodes’ impedance and charge-transfer resistance (Rct) are measured in order to investigate the electrode–electrolyte interfacial features. Fitting and simulation studies were carried out using the modified Rundles circuit (see Figure 2a) as the equivalent circuit. In the equivalent Randle’s circuit, Rs, CPE, Rct, and Zw are the uncompensated solution resistance, constant phase element (which accounts for the bulk capacitance of the catalyst), charge transfer, and mass-transport contribution (Warburg resistance) for ion diffusion, respectively. Rct values were the primary focus of this proof-of-concept study. With this parameter, an indication of the charge-transfer resistance at the electrode surface can be determined. Charge transfer at the electrode interface occurs more efficiently when the diameter of Rct is smaller. Therefore, the smaller the diameter of Rct, the better the conductivity of the electrode interface may be. The Rct values for each type of electrode (bare GPE, Pd/GPE, Pd/GPE, and Pd-Cu/GPE) were 506.3, 186.4, 104.5, and 45.1 Ω, respectively.

When Pd-Cu is added to GPE, the conductivity of the nanocomposite increases 11.2-fold when compared to bare GPE. Also, the Pd-Cu aerogels with the different mole ratios 2.5:2.5 and 1:5 were shown to have a lower intensity than Pd-Cu aerogel with the ratio 5:1. Based on these results, a Pd-Cu aerogel with the ratio 5:1 was chosen as the optimal catalyst.

### 3.3. pH Study

In general, protons play an important role in the electrooxidation of BPA [35,36,37], and the acidity of the electrolyte solution must be optimized. The DPVs for the pH study of BPA in 0.2 M PBS electrolyte and buffer solutions with pH ranged from 3.0 to 8.0 using Pd-Cu/GPE are shown in Figure 3a.

The variation of the oxidation potential of BPA was plotted against pH in Figure 3b. According to this figure, by increasing the solution pH, the anodic peak potential (*E*p) shifts to the negative for BPA. This variation is depicted in Equation (1).
*E*p_(BPA)_ = −0.0525pH + 0.901 (R^2^ = 0.9985) (1)

Based on this equation, the slope value for BPA is 0.0526 V/pH, which is very close to the theoretical value of the 0.0591 V/pH unit (d*E*p = dpH = −2.303 (*mRT/n*F)pH) [38]. The result indicates that the electrochemical redox of BPA at Pd-Cu/GPE should have an equal number of protons and electrons (m = n) in the electrode reactions. The suggested electrooxidation mechanism for BPA is shown in Equation (2) as follows:

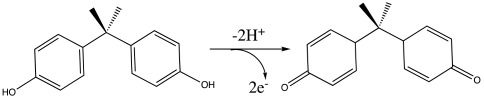
(2)

This result suggested that BPA electrooxidation is a two-electron and two-proton process, which is consistent with other reported works [35,36,37,39,40,41,42,43].

### 3.4. Calibration Curve

The DPVs were used to investigate the relationship between anodic peak currents and concentrations of BSA under optimum conditions, with the results shown in Figure 4a and the calibration curve depictions in Figure 4b. Based on Figure 4b, there are clearly defined increasing oxidation peaks of analytes with the increase in the BSA concentration and two linear segments with variable slopes in the ranges 0.04–85 and 85–305 µM. Based on the two linear segments, it was observed that the Pd-Cu/GPE nanocomposite exhibited different behaviors for BPA at low and high concentrations. In order to explain the observed difference in the slopes of two linear segments, we hypothesized that the surface activity of the Pd-Cu/GPE electrode varies with the analyte concentration. Analyte concentrations and ESA have been reported previously, and they are directly comparable to the current signals. By using Pd-Cu/GPE in this manner, BPA with low concentrations could be detected with a significantly larger effective surface area than that with high concentrations. The slope of the second linear segment of BSA is, thus, reduced at high concentrations of the analyte. A decrease in the diffusion layer thickness and mass transfer limitation is predicted if analyte concentration is increased, according to Bard & Faulkner [44], and Zare et al. [45]; with increasing concentrations, the sensitivity of BSA decreased. This result can be attributed to the kinetic limitations of electron transfer.

In accordance with Equation (3), we calculated the limit of detection (LOD) as follows:(3)$$\mathrm{LOD}=\frac{3S_{bk}}{m}$$
where the *S_bk_* and *m* are the standard deviations of the background signals (n = 10) for Pd-Cu/GPE and the slope of the first segment of the calibration curve, respectively. The theoretical LOD value was 0.02 μM. In the case of the detection of Pd-Cu/GPE, Table 1 presents the figure of merits for comparison of the Pd-Cu/GPE with the literature. We found that the electrochemical performance of Pd-Cu/GPE electrodes is comparable with the recent literature concerning the speciation of BPA. According to Table 1, Pd-Cu/GPE offers a relatively larger linear range and a relatively acceptable LOD for BPA.

### 3.5. Real Sample

BPA is primarily released in wastewater effluents and cosmetic products [57]. Several organic and inorganic chemicals are abundant in real samples. Previous research has been conducted to identify common constituents of domestic wastewater, such as artificial sweeteners, caffeine, pesticides, metal ions, and pharmaceutical compounds [58]. BPA is hazardous to human health and aquatic biota due to its toxicity, bioaccumulation, and non-biodegradability [13]. As a result, it is critical that the proposed sensor be capable of detecting BPA in these real-world samples. The Pd-Cu/GPE was used to evaluate BPA in four different real samples, including domestic wastewater, tap water, bottled water, and commercial hair dye samples, to assess the practical applicability of our electrochemical approach. The samples were diluted in 0.2 M PBS (pH 2.0), and the DPVs were measured using the standard addition technique; the results were confirmed using the HPLC standard method [28]. As shown in Table 2, acceptable recovery values were obtained, indicating the applicability of Pd-Cu/GPE as a promising tool for determining the trace amount of BPA in real samples. Except for commercial hair dye, which contained BPA, BPA was not present in domestic wastewater, tap water, or bottled water samples. BPA was hypothesized to be present in the hair dye sample due to leaching from the plastic container in which the dye was stored. Similar results were reported by Bhargav et al. [33] and Andrianou et al. [59].

## 4. Conclusions

In this proof-of-concept work, a novel Pd-Cu aerogel was synthesized and used as a new modifier in GPE. The Pd-Cu/GPE was used for the electrochemical determination of BPA, and under the optimum condition, the modified electrode improved the peak current of the DPV signal of BPA in comparison to bare GPE, Pd/GPE, and Cu/GPE due to a large redox-active surface area and improved BPA oxidation kinetics. The calibration curve was linear up to 305.0 μM, with a limit of detection of 20 nM. The Pd-Cu/GPE was used for the determination of BPA in domestic wastewater, tap water, bottled water, and commercial hair dye samples, with the recovery values ranging from 97.0 to 104%. Based on this study, the Pd-Cu/GPE is a promising material for development as a miniaturized prescreening device and rapid prescreening tool for detecting BPA in environmental samples at the point of care.

## Figures and Tables

**Figure 1 materials-16-06081-f001:**
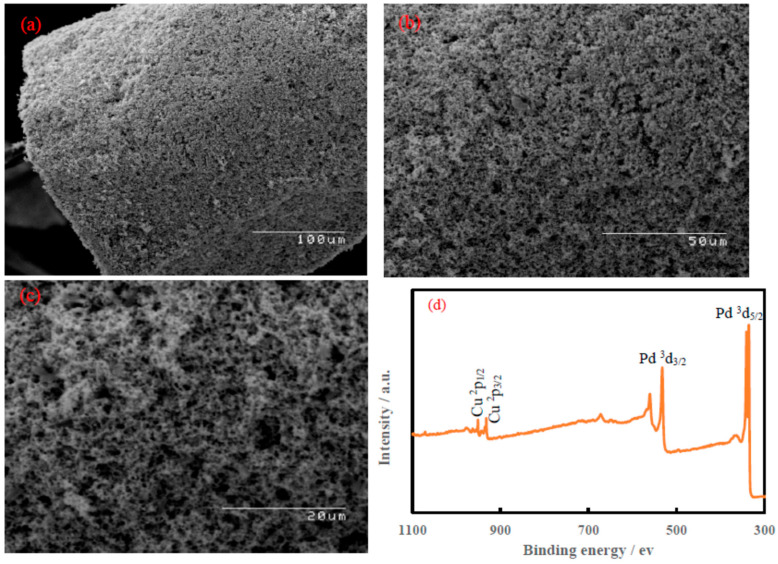
(**a**–**c**) SEM images of Pd-Cu aerogel with different magnifications, (**d**) XPS for Pd-Cu aerogel, (**e**,**f**) TEM images of Pd-Cu aerogel with different magnifications, and (**g**,**h**) AMF images of Pd-Cu aerogel with different magnifications.

**Figure 2 materials-16-06081-f002:**
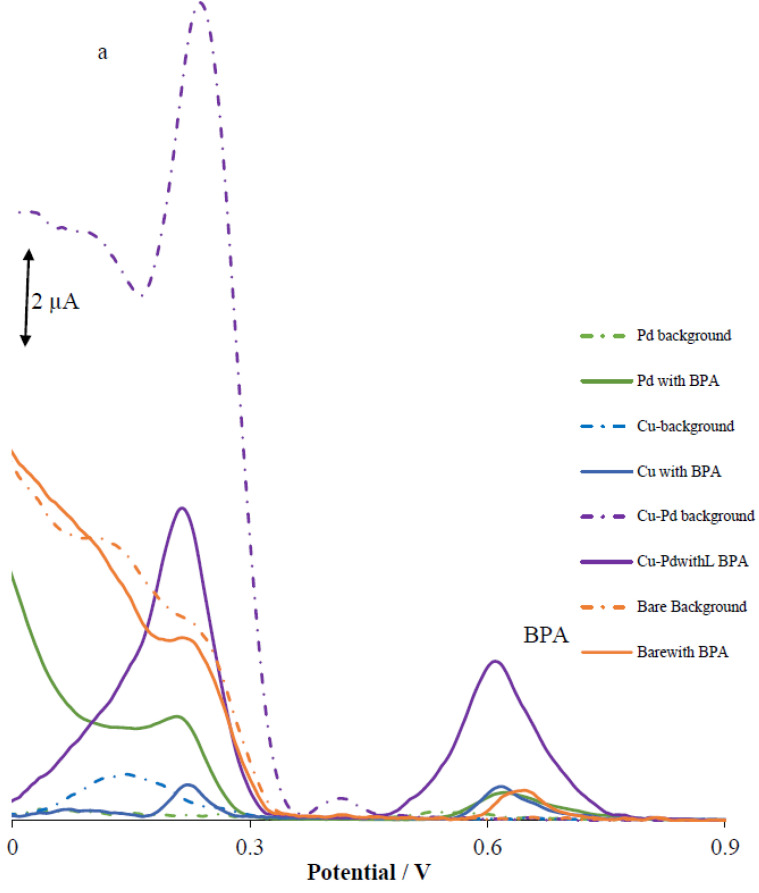
(**a**) DPV in absence and presence of BPA with concentration of 25 µM, using bare GPE, Cu/GPE, Pd/GPE, and Pd-Cu/GPE; (**b**) Nyquist plots for bare GPE, Cu/GPE, Pd/GPE, and Pd-Cu/GPE.

**Figure 3 materials-16-06081-f003:**
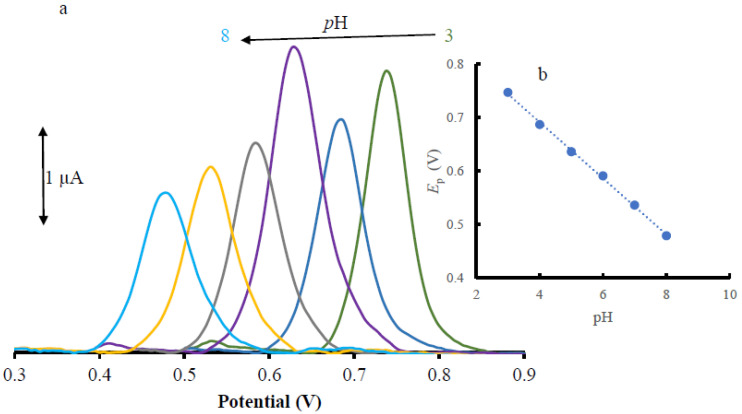
(**a**) DPV of BPA (25.0 µM at the Pd-Cu/GPE in varying pH conditions); (**b**) Plot for the variation of peak potentials of BPA vs. pH.

**Figure 4 materials-16-06081-f004:**
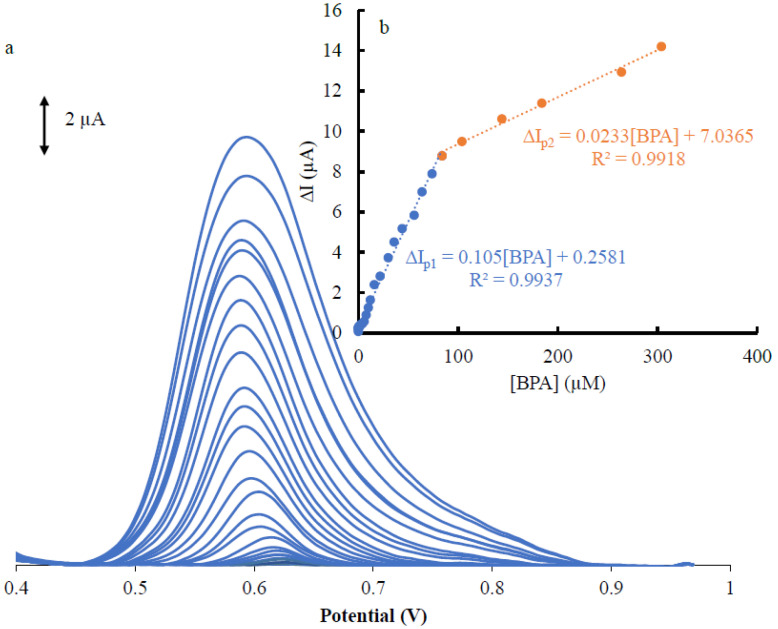
(**a**) DPV for the determination of BPA at Pd-Cu/GPE, (**b**) relationship between increasing anodic peak current signals and the corresponding increasing concentrations for BPA.

**Table 1 materials-16-06081-t001:** Comparison of Pd-Cu/GPE with other modified electrodes in the literature for the determination of BPA.

Electrode	Method	pH	Linear Range (µM)	LOD (nM)	Ref.
CuPc/MWCNT-COOH/PGE ^a^	DPV	2	0.1–27.5	18.9	[46]
MWCNTs/CuFe_2_O_4_/GCE ^b^	DPV	7	0.01–120	3.2	[47]
Carbon felt ^c^	CV	7	0.01–150	480	[48]
Mo_2_Ti_2_AlC3MAX phase/MWCNT/GCE ^d^	DPV	7	0.01–8.5	2.7	[49]
CAS-CB/GCE ^e^	LSV	5	0.49–24	250	[50]
AgNPs/GPUE ^f^	DPV	7.4	2.5–15	240	[51]
CB/AuSNPs/SNGCE ^g^	DPV	7	0.5–15	60	[52]
Ce-doped ZnO/CPE ^h^	DPV	7	0.05–3.2, 3.2–62.0	18	[53]
MNR-PMDAN/GPE ^i^	SWV&DPV	2	0.01–5.0, 5.0–40.0	4.3	[37]
NiNPs/NCN/CS/GCE ^j^	DPV	5	0.1–2.5, 2.5–15	45	[54]
AuNPs/MoS_2_-NFs/IL-graphene/GCE ^k^	DPV	6.5	0.05–0.8, 0.8–4	28	[55]
PCL/GO membranes ^l^	DPV	7.4	0.025–1, 1–20	23	[56]
Cu-Pd/GPE	DPV	5	0.04–85, 85–305	20	This Work

^a^ Pencil graphite electrode modified by multi-walled carbon nanotubes and copper phthalocyanines, ^b^ Glassy carbon electrode modified by multi-walled carbon nanotubes and copper ferrite nanocomposite, ^c^ Carbon felt, ^d^ Glassy carbon electrode modified by multi-walled carbon nanotubes and Mo_2_Ti_2_AlC_3_, ^e^ Electrochemical sensor based on casein and carbon black, ^f^ Pen sensors with graphite-polyurethane and silver nanoparticles composite electrodes, ^g^ Sonogel-Carbon electrode modified by carbon black and gold sononanoparticles, ^h^ Carbon paste electrode modified by Ce-doped ZnO nanorods, ^i^ Fe_3_O_4_ nanorods (MNRs) and a poly(3,4-methylenedioxy)aniline (PMDAN) nanocomposite-modified graphite paste electrode (GPE), ^j^ Glassy carbon electrode modified by three-dimensional hierarchical cylinder-like nickel nanoparticle/nitrogen-doped carbon nanosheet/chitosan nanocomposite, ^k^ Glassy carbon electrode modified by gold nanoparticles-MoS_2_ nanoflowers/ionic liquid-functionalized graphene (AuNPs/MoS_2_-NFs/IL-graphene) nanocomposites, ^l^ Electrospun membranes from polycaprolactone/graphene oxide.

**Table 2 materials-16-06081-t002:** Determination of BPA using Pd-Cu/GPE in domestic wastewater, tap water, bottled water, and commercial hair dye in comparison with standard HPLC method (n = 3).

Sample	Pd-Cu/GPE		HPLC Method				
	Detected (µM)	Spike (µM)	Found ± SD ^a^ (µM)	R ^b^ (%)	Detected (µM)	Found ± SD ^a^ (µM)	R ^b^ (%)
Domestic wastewater	-	10.0	10.2 ± 0.11	102.0	-	9.85 ± 0.08	98.5
Tap water	-	10.0	9.87 ± 0.07	98.7	-	10.4 ± 0.05	104.0
Bottled water	-	10.0	9.92 ± 0.05	99.2	-	10.3 ± 0.10	103.0
1% Commercial hair dye	18.5	5.0	4.89 ± 0.11	97.8	18.9	4.85 ± 0.09	97.0

^a^ SD and ^b^ R denote standard deviation (n = 3) and recovery, respectively.

## Data Availability

Not applicable.

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
