# Peer review of "Palladium–Copper Bimetallic Aerogel as New Modifier for Highly Sensitive Determination of Bisphenol A in Real Samples"

_materials, 2023, doi:10.3390/ma16186081_

Round 1

Reviewer 1 Report

The paper entitled “Palladium-Copper bimetallic aerogel as new modifier for highly sensitive determination of bisphenol A in real samples” is a good attempt. The arrangement of material is reasonable, and all results have been adequately discussed in the manuscript. There are several small errors which can easily be eliminated without going into further experimental work. Some of my specific suggestions are listed hereunder;

1. In the abstract some well-established techniques are given in full, their abbreviated form will be enough.

2. Pay special attention to, Line 50-51, there is no need to explains LC-MS in the way as it is. Line 52, quantitation?, graphite-paste electrode and graphite paste electrode, which his the correct one?

3. All chemical formula of compounds are wrong it terms of subscript position of digits. 

4. Line 88, specify the oxidation state of Cu like Pd and better to use their symbols after first full mention.

5.       Molar quantities along with exact mass are necessary to be mentioned in experimental section.

6. Figure 1 is confusing merge all sub-figures as one and insert. Check its caption, there seem some confusing terms specifically after, (c), (f) and (h).

7. Cu2p3/2 superscript and subscript positions in the respective figure with respect to the text.

8.  There is an extra dot in lines 173-174 omit one to make the sentence continuous.

9. Short sentences like in line 179, can easily be merged in the sentence prior to it. Check the entire text for such cases.

10. Quality of Figure 2 is poor, it is better to disintegrate it into two figures with improved quality. Check text in other figure too, it is blur.

11. Check sentence completeness in line 208. ……….into it GPE.

12. Several sentences start with citation of Figure or Table, must be revised as appropriate.

13. Line 246, Bard and Faulkner et al., and Reference in the list, check and correct. Also add city of publication of the book in references.

14. Revise "Conclusion" with incorporation of numerical data to reflect results of the instant study.

Moderate changes are needed 

Author Response

Response Letter for  Reviewer #1

Comments and Suggestions for Authors

The paper entitled “Palladium-Copper bimetallic aerogel as new modifier for highly sensitive determination of bisphenol A in real samples” is a good attempt. The arrangement of material is reasonable, and all results have been adequately discussed in the manuscript. There are several small errors which can easily be eliminated without going into further experimental work. Some of my specific suggestions are listed hereunder;

  1. In the abstract some well-established techniques are given in full, their abbreviated form will be enough.

R1; We thanks for your comment and the corrections are made in abstract in page 1, line 12.

  1. Pay special attention to, Line 50-51, there is no need to explains LC-MS in the way as it is. Line 52, quantitation?, graphite-paste electrode and graphite paste electrode, which his the correct one?

R2. We appreciate for your comments. We deleted explaining LC-MS in page 2, Lines 52-52. In case of graphite paste electrode, we checked these two formats and changed all of them to “graphite paste electrode” in whole revised manuscript.

  1. All chemical formula of compounds are wrong it terms of subscript position of digits. 

R3. We appreciate your comment, and we have rectified all chemical formulas for compounds where the subscript position existed.

  1. Line 88, specify the oxidation state of Cu like Pd and better to use their symbols after first full mention.

R4. We appreciate your comment and we definite palladium (Pd), copper (Cu)  and palladium- copper (Pd-Cu) in page 2, lines 71, 72, and 78. All oxidation states of  Cu and pd in bimetallic catalyst are zero.

  1. Molar quantities along with exact mass are necessary to be mentioned in experimental section.

R5. We appreciate your comment and the corrections are made in page3, lines 104 and 105.

  1. Figure 1 is confusing merge all sub-figures as one and insert. Check its caption, there seem some confusing terms specifically after, (c), (f) and (h).

R6. Thanks for your comments and we change it the corrections are made in caption of Figure 1 in page 5, lines 164-166.

  1. Cu2p3/2superscript and subscript positions in the respective figure with respect to the text.

R7. We appreciate your comment and the corrections are made in Figure 1d and in the main text in page 5, line 168.

  1. There is an extra dot in lines 173-174 omit one to make the sentence continuous.

R8. We appreciate for your comment and the correction is made in page 5 and line 181.

  1. Short sentences like in line 179, can easily be merged in the sentence prior to it. Check the entire text for such cases.

R.9. We appreciate your comment and short sentence merged in page 5, line 179.

  1. Quality of Figure 2 is poor, it is better to disintegrate it into two figures with improved quality. Check text in other figure too, it is blur.

R.10. We appreciate your comment and the quality of the all figures increased.

  1. Check sentence completeness in line 208. ……….into it GPE.

R.11. We appreciate your comment and the correction is made in Page 7, line 206-207.

“When Pd-Cu is added to GPE, the conductivity of the nanocomposite increases 11.2-fold when compared to bare GPE.”

  1. Several sentences start with citation of Figure or Table, must be revised as appropriate.

R.12. We appreciate for your comment and the corrections are made in whole revised manuscript

  1. Line 246, Bard and Faulkner et al., and Reference in the list, check and correct. Also add city of publication of the book in references.
  2. 13. We appreciate for your comment and the corrections are made in P. 9, line 244 and 245 and in the cities of this reference were added to reference section in page 15, line 430.
  3. Revise "Conclusion" with incorporation of numerical data to reflect results of the instant study.

R.14. We appreciate for your comment and the corrections are made in conclusion in page 11, line 291 and 295-301.

Reviewer 2 Report

The authors in the present manuscript to show that a bimetallic palladium-copper aerogel was synthesized and used for modification of graphite paste electrode (Pd-Cu/GPE) allowing the sensitive determination of bisphenol A (BPA). Scanning electron microscopy, Transmission electron microscopy, X-ray photoelectron spectroscopy, and atomic force microscopy were used for characterization of the Pd-Cu aerogel. To elucidate the properties of the Pd-Cu/GPE, the electrochemistry methods such as differential pulse voltammetry (DPV) and electrochemical impedance spectroscopy were used. DPV measurements were conducted in phosphate electrolyte and buffer solution (0.2 M PBS, pH 5) at a potential range from 0.4 to 0.9 V vs. Ag/AgCl. The DPVs peaks currents increased linearly with BPA concentrations in the 0.04-85 and 85-305 M ranges, with a detection limit of 20 nM. The modified electrode was successfully used in real samples to determine BPA and the results were compared to the standard HPLC method. The results showed that the Pd-Cu/GPE had good selectivity, stability, and sensitivity for BPA determination. The authors should address the following issues and information’s before publication acceptance in the prestigious ‘Materials’ Journal:

1. In Introduction, what is the lowest concentration of BPA in the environment?

2. In Introduction, authors should add a Table that compares the different metals, composites, properties and application values with published literatures?

3. Why did the authors select Pd-Cu catalysts for this study?

4. In Introduction, authors should explain clear novelty of this study? Just provide this statement is not enough “addition of Pd is expected to improve the reducibility of copper and enrich the catalyst surface with electrons, hence increasing the electron transferring rate”. Discuss the depth research gap.

5. On what basic authors decided these two Pd: Cu ratios (1:1 & 1:10)?

6. Authors should calculate particle size using TEM images and perform XRD analysis of samples?

Minor editing of English language required.

Author Response

Response Letter for Reviewer #2
Comments and Suggestions for Authors
The authors in the present manuscript to show that a bimetallic palladium-copper aerogel was synthesized and used for modification of graphite paste electrode (Pd-Cu/GPE) allowing the sensitive determination of bisphenol A (BPA). Scanning electron microscopy, Transmission electron microscopy, X-ray photoelectron spectroscopy, and atomic force microscopy were used for characterization of the Pd-Cu aerogel. To elucidate the properties of the Pd-Cu/GPE, the electrochemistry methods such as differential pulse voltammetry (DPV) and electrochemical impedance spectroscopy were used. DPV measurements were conducted in phosphate electrolyte and buffer solution (0.2 M PBS, pH 5) at a potential range from 0.4 to 0.9 V vs. Ag/AgCl. The DPVs peaks currents increased linearly with BPA concentrations in the 0.04-85 and 85-305 M ranges, with a detection limit of 20 nM. The modified electrode was successfully used in real samples to determine BPA and the results were compared to the standard HPLC method. The results showed that the Pd-Cu/GPE had good selectivity, stability, and sensitivity for BPA determination. The authors should address the following issues and information’s before publication acceptance in the prestigious ‘Materials’ Journal:
1. In Introduction, what is the lowest concentration of BPA in the environment?
R. 1. We appreciate for your comment and we added the following sentence in page 1, lines 37-39 and also a new refrence was added as reference 14 in page 12, lines 343-345.
“The maximum concentration level of BPA in drinking water established by European Union, China and Japan are 2.5 μg/L ( 11 nM), 10.0 μg/L ( 44 nM), and 100 μg/L ( 440 nM) , respectively [14 in revised manuscript].” in page 1, lines 37-39 and reference 14 in page 12, lines 326-327.
[14] Duenas-Moreno, J.; Mora, A.; Cervantes-Aviles, P.; Mahlknecht, J.; Groundwater contamination pathways of phthalates and bisphenol A:origin, characteristics, transport, and fate – A review. Environ. Int., 2022, 170, 107550.
2. In Introduction, authors should add a Table that compares the different metals, composites, properties and application values with published literatures?
R2. We appreciate for your comment. As you know we compared our modified electrode with literature in Table 1 in page 10.
3. Why did the authors select Pd-Cu catalysts for this study?
R3. We appreciate for your comment and the corrections are made in the introduction in page 2, lines 71-84 as follow;
“Palladium (Pd) has been employed in a variety of catalytic applications for fuel electrooxidation [22, 23]. Copper (Cu) nanoparticles has been considered outstanding electrochemical catalysts [24] and copper content of the alloy examined for electrooxidation reaction [25]. Also, addition
of the Pd is expected to improve the reducibility of copper and enrich the catalyst surface with electrons, hence increasing the electron transferring rate [26, 27]. The existing research on Pd-Cu bimetallic nanoparticles is mainly focused on its catalytical capability for the electroreduction of CO2 and NO [26, 27]. According to this research, bimetallic palladium-copper( Pd-Cu) nanoparticles catalyst has the potential advantage of providing binding site variety, which boosts catalytic activity and has remarkable potential in the electrocatalysts research [28]. Mo et al. was reported application of the first bimetallic Au-Pd incorporated in the carboxylic multi-walled carbon nanotubes as supporter to improve electron transport of the poly (diallyldimethylammonium chloride) and used for electrochemical determination of BPA [29]. In this study, different types of metallic aerogels, Pd, Cu and bimetallic aerogels Pd-Cu with three different ratios 5:1, 3:3, and 1:5 were prepared.”
22. A. Shafaei Douk, H. Saravani, M. Noroozifar, K.-H. Kim, Tuning the morphology of Pd aerogels for advanced electrocatalysis of formic acid Microporous and Mesoporous Materi-als 344 (2022) 112206.
23. M. Zareie Yazdan-Abad, M. Noroozifar, A. Shafaei Douk, A. R. Modarresi-Alam, H. Saravani, Shape engineering of palladium aerogels assembled by nanosheets to achieve a high performance electrocatalyst, Applied Catalysis B: Environmental 250 (2019) 242–249.
24. Lin, F.; Jiang, X.; Boreriboon, N.; Wang, Z.; Song, C.; Cen, K. Effects of supports on bimetallic Pd-Cu catalysts for CO2 hydrogenation to methanol. Appl. Catal. A: Gen. 2019, 585, 117210.
25. Z. Khan Ghouri, N. A. M. Barakat, H.Y. Kim, Influence of copper content on the elec-trocatalytic activity toward methanol oxidation of CoχCuy alloy nanoparticles-decorated CNFs, Scientific Reports, 2015, 5,16695
26. Xing, F.; Jeon, J.; Toyao, T.; Shimizu, K.I.; Furukawa, S. A Cu–Pd single-atom alloy catalyst for highly efficient NO reduction. Chem. Sci. 2019, 10(36), 8292–8298.
27. Ashraf, G.; Asif, M.; Aziz, A.; Wang, Z.; Qiu, X.; Huang, Q.; Xiao, F.; Liu, H. Nanocomposites consisting of copper and copper oxide incorporated into MoS4 nanostructures for sensitive voltammetric determination of bisphenol A. Microchim. Acta 2019, 186, 337.
28. Sharma, G.; Kumar, A.; Sharma, S.; Naushad, M.; Dwivedi, R.P.; ALOthman, Z.A.; Mola, G.T. Novel development of nanoparticles to bimetallic nanoparticles and their composites: A review. J. King Saud Univ. Sci. 2019, 31(2), 257–269.
29. Mo, F.; Xie, J.; Wu, T.; Liu, M.; Zhang, Y.; Yao, S. A sensitive electrochemical sensor for bisphenol A on the basis of the AuPd incorporated carboxylic multi-walled carbon nanotubes. Food Chem. 2019, 292, 253-259.
4. In Introduction, authors should explain clear novelty of this study? Just provide this statement is not enough “addition of Pd is expected to improve the reducibility of copper and enrich the catalyst surface with electrons, hence increasing the electron transferring rate”. Discuss the depth research gap.
R4. We appreciate for your comment. The corrections are made in the introduction in last paragraph in page 2, lines 71-84.
5. On what basic authors decided these two Pd: Cu ratios (1:1 & 1:10)?
R5. We appreciate for your comment. We test five different catalyst including Cu, Pd, and Pd:Cu with different mole rations Pd:Cu 5:1, 3:3, 1:5, (Page 3, section 2.2). Based on our results aerogel with ratio 5: 1 showed highest sensitivity. (Page 7, lines 206-209.
6. Authors should calculate particle size using TEM images and perform XRD analysis of samples?
R6. We appreciate for your comment. Our modifier is a Pd-Cu aerogel with porous structure as we shown in Figures 1a-c by SEM and Figures 1e-f by TEM and Figures 1g-h by AFM. Also, the composition of this catalyst was reported by XPS.

Round 2

Reviewer 1 Report

Authors have made all the suggested changes to my entire satisfaction. If authors change the color font inside the figure to make it visible, in some cases, it will make the paper more attractive.  There is no further technical point to hinder me from my positive recommendation in favor of the manuscript.